# EGLNN: Enhanced Graphless Neural Network for IoT Data Storage Transaction Security

## Abstract

With the rise of 5G and the IOT, the amount of data generated by IoT devices has exploded. Ethereum has become a secure tool for storing and trading IoT data due to its openness and tamper-proof nature. However, as Ethereum becomes more and more popular, the Ethereum platform has also become a hotbed for various types of cybercrimes, so ensuring the security of the Ethereum network is crucial. Recently, algorithms based on GNN have been seen as an effective way to detect abnormal nodes in the network. However, through analysis, this work finds that its original network structure is not optimal, directly applied to the existing GNN model with poor results. Meanwhile, it is understood that most of the current GNNs rely on the message-passing principle, which leads to slow model training and inference, and large model size. It is quite challenging to directly apply traditional GNN algorithms in industrial scenarios with limited space and high feedback time requirements. This study proposes a knowledge distillation-based algorithm called Enhanced Graph-Less Neural Network .EGLNN estimates more realistic graph structures through Bayesian graph structure estimator and solves the problem of large-scale GNN models being difficult to be widely applied in industry through the faculty-student distillation method.

## 1 Introduction

The Internet of Things (IoT)Xu et al. (2022) refers to a group or groups of tightly connected devices that form a network through wireless or cable communication technology and work together to the common objectives of its users.With the rapid development of 5G, the amount of data generated by connected devices in the Industrial IoT (IIoT) paradigm has significantly increased within Industry 4.0. This data holds immense value across fields such as technology, economy, energy, and smart cities. An rising number of consumers and businesses see the transactional value of IoT data. Consequently, IoT data trading platforms have extensive applications and promising prospects.Currently, most IoT data trading platforms rely on third parties for data storage and transactionsMohamed & Mohamed (2019). However, this data trading approach is not inherently secure and may lead to data security issues such as industrial information leaks due to platform credibility concerns. Blockchain technologyWu et al. (2019), with its attributes of decentralization, transparency, and immutability, offers a potential solution to these problems. As a result, an increasing number of researchers are exploring the utilization of blockchain technology for storing and trading IoT data without dependence on third-party platformsEsposito et al. (2018).

However, with the increasing popularity of Ethereum, the Ethereum network platform has also become a hotbed of various cybercrimes Wu et al. (2021). Among them, phishing scams are the most harmful among all kinds of cybercrimes Li et al. (2022a).This shows that the security of the Ethereum platform has become a key issue affecting the development of the Ethereum ecosystem. Currently, the most popular method for detecting abnormal nodes in the Ethereum network is to convert the abnormal node detection problem into a node classification problem through network representation learning, and identify abnormal nodes in the network by learning the characteristic information of the network Lin et al. (2020). Wang et al.Wang et al. (2022) presented a heterogeneous network-based network embedding approach to mine implicit information in Ethereum transactions.

In recent research on anomaly detection in the Ethereum network, algorithms applying network representation learning and graph neural networks have made significant progress. However, what we still need to think about is whether this type of GNN model is enough to solve the Ethereum network security problem in the context of industrial information storage transactions? In this work, the topology of the Ethereum transaction network was first analyzed, and it was found that the degree of nodes in the Ethereum transaction network exhibits the characteristics of a long-tail distribution Liu et al. (2020). This shows that the original transaction network structure of Ethereum may not be optimal, making it difficult for the GNN model trained directly using the original graph to obtain optimal network structure information.Secondly, by analyzing the principle of the GNN algorithm, we know that most of the current GNNs rely on the principle of message passing Yang et al. (2023b). This makes the training and execution speed of the GNN model slower and the size of the trained model larger. Although this feature of GNN can guarantee good results in node classification tasks, it is only suitable for scenarios with unlimited memory and speed, and it is not suitable for industrial application scenarios.

After identifying these challenges, this study proposes a model called Enhanced Graphless Neural Network (EGLNN) to Solve Ethereum network security issues in industrial information storage and transaction scenarios. The model's main idea is to transfer a large amount of work from the delay-constrained teacher GNN reasoning to the less time-sensitive student MLP by adopting the method of knowledge distillation (KD) Yang et al. (2023a). The purpose is to transfer the knowledge learned in the teacher model from The typical GNN large model is extracted into a smaller MLP model, such that the student MLP model can perform similarly to the instructor model and has a running speed that the teacher model does not have, so that it can be applied to industrial platforms.

The following are the three main contributions of this paper:

(1) This paper proposes a new GNN model, which gets rid of the scalability and deployment challenges brought about by GNN's data dependence in industrial environments through knowledge distillation, so that it can be deployed to applications that require fast reasoning. in latency-limited applications.

(2) Through research, it was discovered that the original Ethereum transaction network structure is not reliable. This work optimizes the graph structure based on Bayesian reasoning, and replaces the initial node features with position encoding (PE) vectors to ensure that the knowledge transferred by the teacher model to students only contains optimized graph structure information.

(3) Extensive experiments are conducted on the Ethereum dataset collected in this work, and the experimental results show that EGLNN has better performance compared with state-of-the-art methods.

## 2 RELATED WORK

This section first briefly reviews two techniques related to this work – graph neural network and knowledge distillation, and introduces related work.

### 2.1 GRAPH NEURAL NETWORK

Current graphic neural network algorithms can be roughly divided into the following five categories: graphical network-based (GCN), graphical attention network (GAT), graphic self-coding (GAE) based, graphic generating model (GAN) based and graphical pooled neural networks (GPN) based.

GCN Kipf & Welling (2016) is one of the most classic and basic algorithms in graphic neural networks, which updates the characteristic vector of each node by aggregating the characteristics of neighbouring nodes . Chen et al.Chen et al. (2019) proposed GIN, which is a graph convolutional neural network based on graph isomorphism. It constructs the embedding vector of nodes by accumulating and splicing the feature vectors of neighboring nodes. The algorithm based on GAT Perozzi et al. (2014) is a graph neural network algorithm based on attention mechanism. Different from GCN, GAT can assign different weights to each neighbor node, and weight the contributions of different neighbor nodes when computing node feature vectors. GATv2 Brody et al. (2021) is an improved version based on variational dropout, which introduces a variational dropout mechanism based on Gaussian noise, which can improve the generalization ability of the model. The algorithm

based on graph autoencoder (GAE) Schulman et al. (2015) is an algorithm that utilizes autoencoder structure to learn graph representation. It treats the graph structure as input, and learns the low-dimensional representation of the graph through the process of encoding and decoding, so as to realize tasks such as graph classification and clustering. An algorithm based on a graph generative model (GAN) Creswell et al. (2018) is a neural network model capable of generating data similar to the input data. In graph neural networks, GANs can be used to generate graph structures that meet specific property or structural requirements. Graph Pooling Neural Network (GPN) Gao et al. (2021) is a GNN based on adaptive graph pooling, which can reduce the size of the graph, improve computational efficiency, and enhance the model's generalization efficacy. It is widely used in graph classification, Graph generation etc.

In summary, most current graph neural network algorithms still directly take the original topological graph as input without considering the incompleteness of the graph structure, which greatly hamper their performance in subsequent tasks.

## 2.2 KNOWLEDGE DISTILLATION

Traditionally, large neural networks need to be run on GPU or TPU to achieve good performance, but large neural networks cannot be run directly on industrial IoT devices with limited computing resources and storage space.Therefore, research on GNN reasoning acceleration has attracted increasing attention.Hinton et al. Hinton et al. (2015) first proposed a knowledge distillation(KD) algorithm.Its central concept is to extract the teacher model's knowledge from a typical large model into a smaller one. In recent years, there are many related studies on knowledge distillation. For example, Zhang et al. Zheng et al. (2021) proposed a model called Cold brew to solve the node long-tail distribution problem by distilling the structural embedding SE learned by the teacher GNN into the student MLP, and Huo et al. Huo et al. (2023) proposed a The double distillation mode enhances the ability of the student model by distilling topological features and attribute features separately.

There are many advantages of knowledge distillation: 1) It can compress the size of the model and compress the knowledge of the large model into a small model while retaining the accuracy of the large model. 2) It can speed up the inference speed. 3) It can improve the model generalization ability. In short, the knowledge distillation algorithm can solve the problem of deploying the GNN algorithm on the Industrial Internet of Things very well. It can greatly reduce the computing and storage costs of the deep neural network, improve the efficiency and performance of the model, and accelerate the deployment and operation on various devices.

## 2.3 ANOMALY DETECTION ALGORITHMS ON THE ETHEREUM NETWORK

The Ethereum ecosystem is gravely threatened by malicious accounts on the Ethereum network.This section summarizes the Ethereum network anomaly detection algorithms released in recent years and introduces their principles and characteristics.Wu et al.Wu et al. (2020) proposed an algorithm called Trans2vec, which is an improvement of Node2vecGrover & Leskovec (2016), is a method for detecting phishing scams by mining transaction records in the Ethereum network. Li et al.Li et al. (2022b) proposed a method called TTAGN to model the temporal relationship in historical transaction records between nodes. This method combines transaction features with common statistical and structural features obtained through graph neural networks to identify phishing addresses.Liu et alLiu et al. (2023) proposed an algorithm called AMBGAT, which enhances the Ethereum network structure by using Bayesian estimation to improve the identification accuracy of phishing nodes.

To sum up, Most of the current Ethereum anomaly detection algorithms are based on graph neural network methods. Although these algorithms have high accuracy, they rely on the aggregation of neighbor nodes that are more than hops away from the target. Therefore, An industrial setting would burden latency-first applications, making it difficult to deploy into latency-bound applications that require fast inference

# 3 THE PROPOSED FRAMEWORK

This section will delve into the methodological framework proposed in this work, which is a novel GNN framework based on knowledge distillation. The method mainly includes two modules:

teacher GNN module and student MLP module. The details are shown in Figure 2 below. In the teacher model, in order to solve the problems of poor topology, long-tail node degree distribution and poor homogeneity of the Ethereum transaction network pointed out in this article, a more realistic graph structure is first estimated through Bayesian inference. It is used for downstream tasks, and then the teacher GNN learns a vector containing only the structural feature embedding (SFE) of the optimal graph based on the optimal graph obtained by Bayesian inference and passes it to the student through the knowledge distillation method. Model MLP1 enables MLP1 to generate an embedding similar to the optimal graph structure feature embedding (SFE) vector by inputting only the node features of the original graph. Next, the structure generated by student MLP1 is used in the student model through self-attention. The feature embeddings and node attribute feature embeddings generated by MLP2 are fused for the final anomaly detection task.

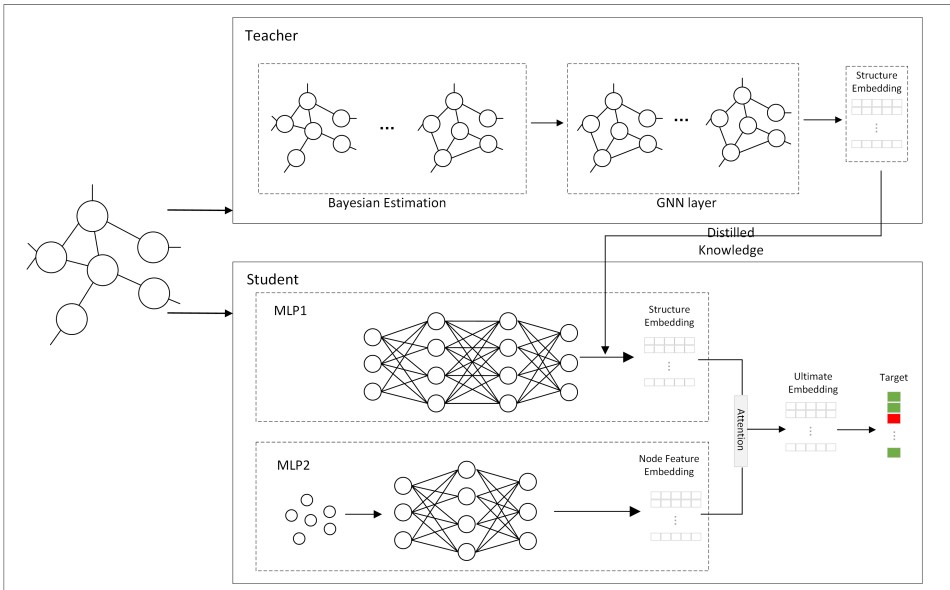

Figure 1: EGLNN Overall Framework

## 3.1 TEACHER GNN MODULE

The main goal of the teacher GNN is to learn an embedding with node structural features and pass it to the student MLP model through distillation. In this way, the student model, given only node features as input, is able to generate embeddings similar to the topological feature embeddings of the optimal graph. This process is designed to enable the student model to obtain topological information from the estimated optimal graph learned by the teacher model and maintain similar features when generating embeddings.

To effectively convey the topological information from the network to the student model, while minimizing the influence of node attribute characteristics, this work uses PE vectors to replace the initial node features. Positional encoding vectors can effectively model positional information in sequence data. In this work, PE vectors are designed to further extract the position information of nodes in the graph structure. One-hot encoding is used to calculate PE vectors, and the dimensions of the vectors are converted to the same dimensions as the initial node features through linear transformation to improve scalability in large-scale graph data. The PE vector of node v can be expressed as:

$$PE_v = W \cdot x_v + b \tag{1}$$

where $W$ is the learnable parameter matrix, $b$ is the bias vector, and $x_v$ is the one-hot encoding of node $v$.

The optimal graph with node attribute features replaced by PEs is then input into a multi-layer GNN network to obtain structural feature embedding (SFE) for distillation.

$$Z_{SFE} = f_{GNN}(Q, PE) \tag{2}$$

## 3.2 STUDENT MODULE

In this work, the student model is designed to consist of two MLP modules. The first student MLP uses the knowledge distillation method to imitate the teacher GNN to generate the structural feature embedding of each node. The second student MLP learns attribute feature embeddings for each node. Then, through the self-attention algorithm, the structural feature embedding generated by student MLP1 and the node attribute feature embedding generated by MLP2 are fused to generate the final embedding for subsequent anomaly detection tasks.

The objective of the first student MLP is to learn the mapping from the input node feature $X$ to $SFE_{Stu}$ through the knowledge distillation method. For node $v$:

$$\hat{e}_v = \xi_1(x_v) \tag{3}$$

The distillation loss is:

$$L = \lambda \sum_{v \in V^L} L_{label}(\hat{y}_v, y_v) + (1 - \lambda) \sum_{v \in V} L_{Distill}(\hat{e}_v, e_v) \tag{4}$$

where $\hat{e}_v$ is the structural embedding generated by student MLP1 under the guidance of the structural feature embedding $e_v$ passed by the teacher GNN, $\hat{y}_v$ is the label prediction of node v by student MLP1, $y_v$ is the true label of node $v$, and $\lambda$ is the balance of two Loss hyperparameters. Continuously reducing the distillation loss $L$ through supervised learning makes the student MLP1 capable of outputting structural feature embeddings similar to the teacher GNN under the condition of only inputting node features.

The second student MLP learns an embedding that contains node attribute feature information by inputting the attribute features $x_v$ of the node:

$$\hat{a}_v = \xi_2(x_v) \tag{5}$$

Considering that the label information of Ethereum nodes may be related to one or several feature information among them, in order to better integrate these two parts of information and extract the information related to Ethereum node labels in these two parts of features, we use adaptive Node feature fusion technology automatically selects the information of the two types of information that is more important to downstream tasks to generate the final node feature embedding.

The fusion process uses the attention adaptive mechanism to automatically learn the importance of different embedded information to the Ethereum fishing node identification task, that is, for the feature information $[\hat{e}_v, \hat{a}_v]$, learn the importance coefficient $[q_e, q_a] \in R^{n*1}$. Specifically, taking node $i$ as an example, we first perform a nonlinear change on its feature vector, and then multiply it by the shared attention vector $\omega$ to obtain its attention value $[q_e^i, q_a^i]$:

$$q_x^i = \omega^T \cdot tanh\left(W \cdot (h_x^i)^T + b\right) \tag{6}$$

where $h_x^i \in R^{1 \times d}$ is one of the two feature embeddings of node $i$, $W \in R^{d' \times d}$ is the trainable weight matrix, $b \in R^{d' \times 1}$ is the paranoia parameter, $\omega \in R^{d' \times 1}$.

Then we use the softmax function to normalize the attention values $q_e, q_a$ to get the final weight:

$$q_e^i = softmax(q_e^i) = \frac{exp(q_e^i)}{exp(q_e^i) + exp(q_a^i)} \tag{7}$$

.

The final feature vector is obtained by combining the two learnt weight coefficients with the corresponding feature information:

$$H^i = q_e^i \cdot h_e^i + q_a^i \cdot h_a^i \tag{8}$$

## 4 EXPERIMENT

In this section, extensive experimental tests will be conducted using the Ethernet transaction dataset collected in this paper to evaluate the effectiveness of EGLNN in performing anomaly detection tasks in Ethernet transaction networks. First, the experimental setup, including the dataset, baseline methodology, and implementation details, is discussed. Then, EGLNN is compared with the baseline approach to evaluate its performance advantages. Next, ablation experimental analysis and model hyperparameter experimental analysis are performed. Finally, the analysis validates the superiority of EGLNN deployment in an industrial environment. It is important to note that the goal of this work is not to pursue the best accuracy, but to improve the model's adaptability in industrial environments as much as possible. It is important to note that the goal of this work is not to pursue the best accuracy rate, but to improve the model's adaptability and scalability in industrial environments as much as possible.

### 4.1 EXPERIMENT SETTINGS

#### 4.1.1 DATA SET

In this paper, we conduct experiments using the dataset collected in subsection 3.2. The collected data is modeled as an Ethernet transaction network, and the Ethernet anomaly detection task is transformed into a graph node classification task. By categorizing the obtained nodes in the Ethernet transaction network, it is possible to effectively identify anomalous accounts in Ethernet transactions and storage, thus improving the transparency and security of IoT storage and transaction platforms. This work employs a typical semi-supervised learning approach that uses both labeled and unlabeled data during model training. The advantage of this approach is that it is able to achieve performance comparable to supervised learning while using less labeled data. With this semi-supervised learning approach, the unlabeled data in the dataset can be fully utilized to improve the generalization ability and performance of the model.

In order to provide a comprehensive evaluation of the approach proposed in this work, the training set is partitioned through three different methods. Specifically, the total dataset is divided into three training sets, D1, D2 and D3, which are used to test the performance of EGLNN. In these datasets, the training set for each type of account contained 60, 80, and 100 randomly assigned labeled nodes of each type, respectively, and the test set was specified to contain 1000 labeled nodes. By using training sets of different sizes and test sets of the same size, it is possible to accurately compare the performance of the models in different data situations and draw more reliable conclusions.

### 4.2 NODE CLASSIFICATION

In this section the semi-supervised node classification performance of EGLNN is evaluated according to the state-of-the-art baseline, and in Table 3, the results of the precision, recall, and F1-score averaged over five independent trials of each method using different random seeds under different test validation sets are reported.

The results of evaluating the efficacy of EGLNN for semi-supervised node classification on the Ethernet transaction network are presented in Table 3. The following observations can be drawn from the results in Table 3:

(1) Compared with previous knowledge distillation methods, the EGLNN method proposed in this paper achieves significant improvements on three different training sets, especially on D1, when the training data is more limited, and the performance improvement is most obvious compared with other knowledge distillation methods. This suggests that the embeddings learned by augmenting the topological features of the graph via Bayesian enhancers in the teacher GNN are more effective relative to the embeddings delivered to the student model by traditional knowledge distillation methods.

Table 1: Node classification results of different methods

| Method | Dataset | D1 | | | D2 | | | D3 | | |
|---|---|---|---|---|---|---|---|---|---|---|
| | Metric | Pre | Recall | F1 | Pre | Recall | F1 | Pre | Recall | F1 |
| traditional method | DeepWalk | 69.30 | 69.31 | 69.30 | 70.63 | 71.66 | 71.14 | 71.60 | 71.54 | 71.57 |
| | GCN | 70.18 | 71.07 | 70.62 | 71.94 | 71.63 | 71.78 | 71.41 | 71.85 | 71.63 |
| Graph Structure Learning method | GEN | 71.55 | 71.84 | 71.69 | 73.43 | 72.62 | 73.02 | 75.5 | 76.97 | 76.23 |
| knowledge distillation method | DistillGCN | 69.10 | 68.11 | 68.60 | 71.11 | 72.65 | 71.87 | 73.10 | 73.12 | 73.11 |
| | T2-GNN | 70.81 | 70.11 | 70.46 | 74.11 | 73.10 | 73.60 | 76.12 | 75.31 | 75.71 |
| Blockchain method | Trans2vec | 77.80 | 76.66 | 77.23 | 81.40 | 81.42 | 81.41 | 82.71 | 81.72 | 82.21 |
| | AMBGAT | 79.40 | 78.81 | 79.10 | 81.80 | 79.79 | 80.78 | 85.31 | **85.70** | **85.10** |
| | EGLNN | **81.07** | **79.96** | **80.51** | **83.40** | **82.30** | **82.85** | **85.46** | 83.94 | 84.70 |

These enhanced embeddings contained richer information and had a more positive impact on the instruction of the student model. This finding emphasizes the superiority of the EGLNN method in knowledge distillation, especially in the case of data scarcity, where the performance enhancement is significant.

(2) The EGLNN method proposed in this paper achieves a significant improvement in the task of anomaly detection in the Industrial Internet of Things (IoT) as compared to traditional methods. Although knowledge distillation is essentially a method of compressing a large model into a small model to improve efficiency, the experimental results in this paper show that the distillation method designed in this study can effectively transfer the topological knowledge learned from the teacher's GNN model corresponding to the real-world situation to the student's model, which solves the problem of the missing original graph structure in a more optimal way and resolves the problem of the large scale of the traditional GNN model that cannot be be applied to industrial scenarios. By fully utilizing the ground truth topological information learned by the teacher model, the student model is able to better learn and represent the graph structure features, thus improving the accuracy and robustness of the model.

(3) Relative to existing neural network-based IoT and Ethernet anomaly detection algorithms, EGLNN can achieve similar or even slightly higher performance, while the performance of previous knowledge distillation methods is significantly lower than them. This result suggests that by adopting Bayesian inference in the teacher GNN to learn the optimal graph, and by replacing the original node attribute information with location-encoded information to generate the structural feature information of the optimal graph and transferring it to the student model, the node categorization ability of the student model can be significantly improved, resulting in the student model to exhibit better performance. This finding emphasizes the superior performance of the EGLNN approach in the industrial IoT anomaly detection task, which is able to achieve better performance while occupying a smaller scale and taking less time compared to traditional anomaly detection algorithms.

### 4.3 ABLATION ANALYSIS

In order to verify the rationality and effectiveness of the model, this work conducts comparative experiments on the Ethereum transaction dataset, comparing EGLNN and its three versions. Specifically, this work tests the following versions of EGLNN separately:

EGLNN-E: Remove the Bayesian structural enhancement module, and only transfer the topological feature information of the original graph through knowledge distillation.

EGLNN-T: Remove the student MLP module and only use the teacher GNN to detect anomalous nodes in Ethereum.

EGLNN-A: Removing the attention fusion module in the student MLP for anomaly detection tasks using only information distilled from the teacher GNN.

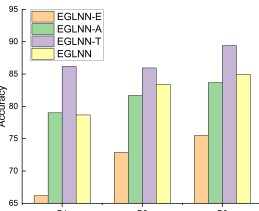 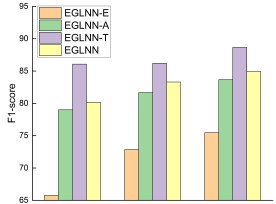 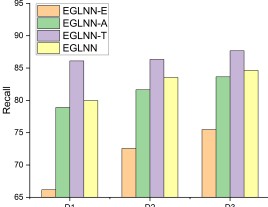

Figure 2: Ablation experimental results of EGLNN.

The results in Fig. 4 show that:

(1) EGLNN outperforms EGLNN-E, which suggests that the model learned by training directly using the raw data map is quite poor. It shows that the structure of the original data map is indeed unsatisfactory and further demonstrates the usefulness of the Bayesian enhancer used in this paper. By applying the Bayesian enhancer to the graph in the teacher GNN, a more complete and realistic graph structure can be provided for subsequent tasks.

(2) The performance of EGLNN is similar to that of EGLNN-T (a module using only the teacher GNN), which indicates that the knowledge distillation algorithm proposed in this paper is able to realize the knowledge migration from the teacher model to the student model in a more effective and comprehensive way. This advantage enables the EGLNN to achieve higher accuracy at a smaller scale, which makes it perfect for anomaly detection tasks in the field of industrial IoT.

(3) EGLNN outperforms EGLNN-A, which indicates that the adaptive algorithm is able to merge the topological features distilled from the teacher model and the node attribute features learned from the student model in a better way compared to the traditional GNN. The introduction of the adaptive algorithm improves the model's performance in detecting anomalous nodes. This finding emphasizes the effectiveness of the adaptive algorithm in fusing knowledge distillation information and student model features, and its improved performance in detecting anomalous nodes.

In summary, the results in Figure 4 validate the soundness and effectiveness of the EGLNN approach. These comparative experiments validate the rationality of the model and reveal the importance and effectiveness of each module in the anomaly detection task.

## 4.4 PARAMETER SENSITIVITY ANALYSIS

In order to deeply investigate the impact of different parameters on model performance, this study evaluates and analyzes a series of parameters on the performance of EGLNN for node classification task on different Ethernet data subsets. When specific parameters are evaluated, all other parameters are set to default values.

First, the effect of embedding dimension on classification performance was evaluated. The classification effectiveness of EGLNN was tested with the node embedding dimensions set to 2, 4, 8, 16, 32, and 64, respectively, and the final classification results are presented in Figure 5.

By observing the experimental results in Fig. 5, it can be learned that the performance of EGLNN on the three datasets peaks when the embedding dimension is 32, while the performance of EGLNN on all three different datasets decreases when the embedding dimension is increased to 64. Therefore, it was chosen to set the embedding dimension to 32 in this study.

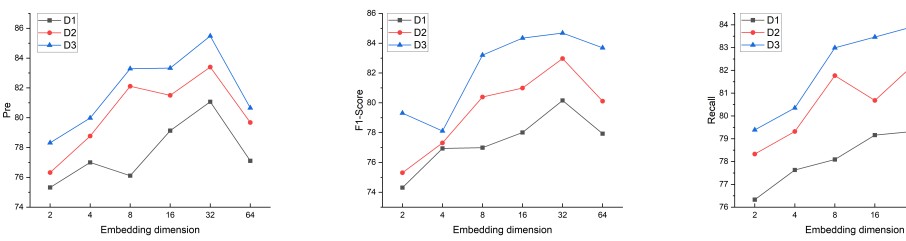

Figure 3: Experimental Results Graph for Variation of Embedding Dimension Parameters Across Different Datasets.

In addition, the equilibrium parameter $\lambda$ for distillation loss was also evaluated in this work, and the classification effectiveness of EGLNN on these three different datasets was tested by varying $\lambda$ from 0.1 to 0.9 in steps of 0.1, and the final classification results are presented in Fig. 6.

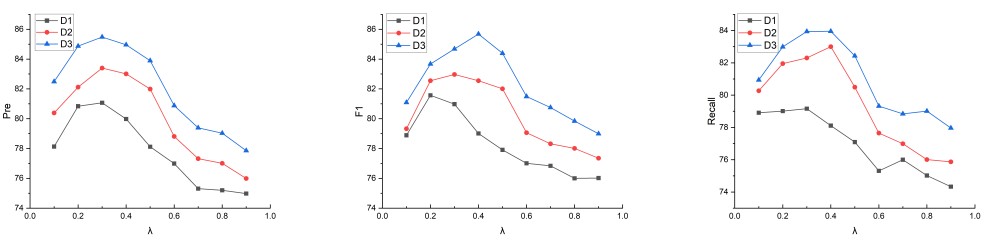

Figure 4: Experimental results graph for variation of balancing parameter $\lambda$ across different datasets.

By observing Fig. 6, it can be noticed that the performance of EGLNN shows a trend of increasing and then decreasing with the increase of $\lambda$. Overall when $\lambda$ takes the value of 0.3, the performance reaches the optimum level. Therefore, this study chose to set the equilibrium parameter for distillation loss to 0.4.

### 4.5 Performance analysis

In order to analyze and validate the superiority of EGLNN deployment in an industrial environment. This work compares EGLNN with several state-of-the-art GNN algorithms to examine their performance under different number of training iterations (where EGLNN-T is the teacher model of EGLNN). The experiments were performed using the same hardware environment and Ethernet transaction dataset, and the same anomaly detection task was performed for each algorithm. The detailed experimental results are shown in Fig. 7.

The experimental results show that:

(1) By using the knowledge distillation algorithm, the EGLNN algorithm performs well in terms of execution speed and occupied resources. Compared with advanced GNN algorithms, EGLNN is able to accomplish the same scale of tasks under the condition of occupying less computational resources and with superior performance, thus saving a lot of computational resources and time. Therefore, EGLNN is more suitable to be applied to anomaly detection tasks in industrial environments where resources and time are limited.

(2) Compared to advanced graph structure learning algorithms, by combining Bayesian inference with knowledge distillation algorithms, EGLNN shows higher efficiency and scalability in processing large-scale graph data. It is only able to better capture the features and topology of real data graphs, but also able to overcome the data-dependency problem of GNN algorithms, effectively utilize the computational resources, and complete the task at a faster speed while ensuring high

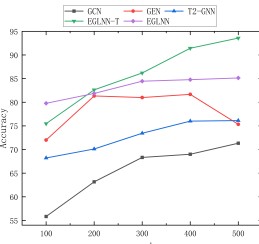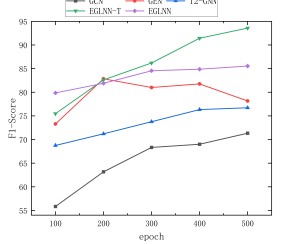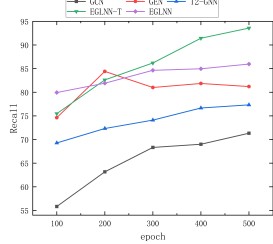

Figure 5: Performance comparison of EGLNN and other algorithms.

performance. This finding emphasizes the advantages of EGLNN in dealing with large-scale graph data, especially in improving efficiency and scalability while maintaining performance.

In summary, by combining the Bayesian graph structure learning algorithm with the knowledge distillation algorithm, EGLNN achieves significant advantages in the industrial IoT anomaly detection task. Its efficient graph structure learning ability, model compression and knowledge migration ability, and high performance in processing large-scale graph data make EGLNN more effective in processing industrial IoT graph data compared with other graph neural network algorithms.

## 5 CONCLUSION

This work employs the algorithm of graph neural networks to detect anomalous nodes in Ethernet networks used for IoT data storage transactions. Different from traditional IoT anomaly detection algorithms, this study innovatively introduces a knowledge distillation algorithm into the traditional GNN algorithm. Through this approach, a more lightweight model that occupies less storage space is successfully trained to solve the problem that large-scale GNN models are difficult to be widely applied in industry. Specifically, this work proposes a model named EGLNN, which consists of a teacher module and a student module. In the teacher module, a more realistic graph structure is first estimated by a Bayesian inference-based approach, and then the structural feature embedding (SFE) of the optimal graph is learned by replacing the attribute features of the nodes with the PE of the nodes on the premise of the estimated optimal graph. Next, the structural feature embeddings learned by the teacher model are transferred as knowledge to the student model through knowledge distillation, so that the student model has similar capabilities as the teacher model. Finally, the structural embeddings learned by the student model are fused with the attribute embeddings using the attention mechanism to obtain the most favorable embeddings for the subsequent anomaly detection task. The effectiveness of the EGLNN was confirmed through extensive experiments and verified the greater advantages of the model over traditional GNN models for industrial IoT anomaly detection tasks. Future work should study the dynamics and heterogeneity of IoT data storage transactions in more depth, and extend EGLNN to dynamic networks containing time-series information, in order to better adapt to the needs of real industrial scenarios.

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

# A APPENDIX

## A.1 PRELIMINARY KNOWLEDGE

To support the proposed EGLNN model, this section provides an overview of the proposed problem, introduces the collection and network analysis of the Ethereum dataset for IoT storage transactions, and briefly outlines the relevant prior knowledge.

### A.1.1 PROBLEM DESCRIPTION

The focus of this paper is the task of detecting abnormal accounts in IoT storage transactions, which can be regarded as a graph node classification problem. The IoT storage transactions studied in this article are conducted on Ethereum. The transaction data set of Ethereum can be represented as a graph $G = (A, X)$, where $A \in R^{n*n}$ is the adjacency matrix representation of the Ethereum transaction network, $n$ is the number of nodes. If $A[i][j] = 1$, it indicates that node $i$ is connected to node $j$. In contrast, if $A[i][j] = 0$, there is no connection between nodes i and j. $X = [x_1, x_2, \ldots x_n] \in R^{n*c}$ is the attribute feature matrix of the node, where c represents the dimension of the node attribute feature, and $x_i$ is the attribute feature vector for the node $i$. Since only a few nodes have label information, this research assignment is a semi-supervised node classification task.

### A.1.2 NETWORK TOPOLOGY ANALYSIS

In order to perform the task of detecting abnormal accounts in IoT storage transactions, enough data is needed to support it. Only with a sufficient amount of training data can the model discover feature relationships and eventually achieve improved classification performance. Due to the transparency and openness of the Ethereum platform, all Ethereum transaction records are accessible. The specific data set collection method follows previous work [57], and the collected network has a total of 376,759 nodes and 1,048,576 edges.

After completing the collection of the data set, this work analyzed the collected Ethereum network topology, including the average degree index, average path length and homogeneity of the nodes in the network (The homogeneity coefficient is used to measure whether nodes in the network like to interact with other nodes with the same label, and its maximum value is 1) analysis. The analysis results show that most nodes in this network have low degrees, transactions between nodes are in their own small worlds and most nodes with the same label do not tend to be connected to each other. However, the performance of graph neural network (GNN) relies heavily on the information of network topology. Although this network has certain topological characteristics, it still has a big gap compared with other networks. It is difficult to directly promote most graph neural network algorithms to the Ethereum transaction network. If the network is directly used with the GNN algorithm will lead to poor model performance. Therefore, it is necessary to estimate a realistic topology through analysis.

### A.1.3 GNN

The graph neural network (GNN) algorithm can encode each node into an embedding vector by iteratively aggregating neighbor information, that is, a message passing mechanism. This message passing mechanism makes the GNN algorithm show outstanding performance in processing various analysis tasks of graph-structured data. Powerful capabilities. where the representation $h_u$ of each node $u$ is iteratively updated in each layer by collecting messages from its neighbors. In the GNN learning node representation procedure, the expression for neighborhood aggregation in the l-th layer of the graph convolution network is:

$$x_v^{(l)} = Prop^{(l)}(h_v^{(l-1)}), v \in N(u) \tag{9}$$

$$h_u^{(l)} = AGGR^{(l)}(h_u^{(l-1)}, \left\{ x_v^{(l)} : v \in N(u) \right\}) \tag{10}$$

Where $N(u)$ is the neighbor set of node $u$, $Prop^{(l)}(\cdot)$ is the node representation generated by aggregating the previous layer of information in the message passing process, $AGGR^{(l)}(\cdot)$ indicates aggregating the information of its neighbors and itself to produce the final node representation. Given the input features $x_0$ of a node $u$ and its neighbors $N(u)$, the final embedding representation of the node can be obtained using Equation (1) and Equation (2).

### A.1.4 KNOWLEDGE DISTILLATION

Knowledge distillation is a model compression method. The goal is to extract knowledge from cumbersome teacher models into lightweight student models, enabling students to maintain similar performance to teachers. This method smoothes the teacher's output by setting a higher temperature in the softmax function so that it contains information about the relationship between classes. The loss function of distillation during the distillation process is weighted by distill loss and student loss.

$$L = \alpha L_{distill} + \beta L_{student} \tag{11}$$

Where $L_{distill}$ is the cross entropy of the student's softmax output results under the same temperature conditions and the teacher model's results is the first part of the Loss function.

$$L_{distill} = -\sum_j^N p_j^T log(q_j^T) \tag{12}$$

$$p_j^T = \frac{exp(v_j/T)}{\sum_k^N exp(v_k/T)} \tag{13}$$

$$q_j^T = \frac{exp(z_j/T)}{\sum_k^N exp(z_k/T)} \tag{14}$$

$L_{student}$ is the cross entropy between the softmax output of the student model and the true label under the condition of T=1.

$$L_{student} = -\sum_j^N c_j log(q_j^1) \tag{15}$$

$$q_i^1 = \frac{exp(z_j)}{\sum_k^N exp(z_k)} \tag{16}$$

## A.2 BAYESIAN INFERENCE

The goal of this subsection is to estimate realistic graphs by constructing a Bayesian-based probabilistic method. When performing graph structure estimation, in order to minimize the bias, in this work, we compile the embedding model to aggregate neighborhood feature information to generate accurate node embedding data into the Bayesian estimator to explicitly constrain the generation of the graph, and in order to reduce the estimation bias To allow the Bayesian inferrer to observe more information, an observation model containing multi-order neighborhood similarity is introduced and injected into the Bayesian inference model to provide node local to global information and constraint estimation The underlying structure of the graph is generated.

Specifically, the original graph G and the node feature matrix The node representation of each layer constructs an observation graph of each layer to describe the similarity of neighborhoods in different levels. The observation graph is calculated in the form of KNN proximity graph $O_i$, and the K proximity graph generated by each layer is composed of the original graph G. The observation set $O = \{G, O_1, O_2, ...O_l\}$, which reflects the optimal graph structure from different views, can be integrated to infer a more reliable graph structure. Finally, the observation set $O$, the output $Z$ of GAT, and the real label $Y$ are put into the Bayesian graph structure estimator, and a more realistic estimated graph $S$ is estimated and inferred by integrating the information provided from different angles. Finally, The estimated map $S$ is fed back to GAT to perform the next round of iteration, and the estimated map $S$ is made more consistent with the actual situation through continuous iterative optimization. Figure 3 shows the entire Bayesian inference framework. By using the Bayesian-based graph structure estimation method, the graph structure in the original Ethereum transaction network can be optimized, filling in the edge data that may be lost during the data acquisition and graph model construction process, and eliminating possible false edges. Obtain a more realistic graph structure and provide more accurate input for subsequent graph neural network algorithms.

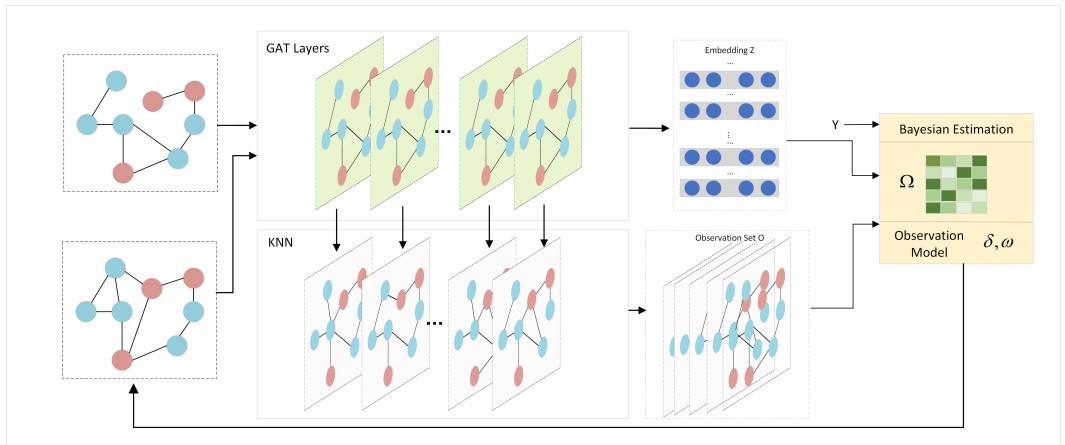

Figure 6: Bayesian Teacher GNN Inference Model

*(a)Embedded model* This part of the embedding model uses the GAT model. When learning the neighborhood characteristics of nodes, the GAT module can dynamically assign different weights to nodes in different neighborhoods without knowing the entire graph structure in advance. This means that it is able to calculate correlations between nodes on a node-by-node basis without the constraints of knowing the complete graph structure in advance.Given the original graph $G = (A, X)$, $A \in R^{n*n}$ is the adjacency matrix representation of the Ethereum transaction network, and $X = \{x_1, x_2, \ldots x_i \ldots x_n\} \in R^{n*c}$ is the attribute feature matrix of the node, where $c$ represents the dimension of the node attribute feature, $x_i$ represents the attribute feature vector of node $i$. The normalized attention mechanism of the GAT model used in this article is:

$$\alpha_{i,j}^l = \frac{exp(LeakyReLU(\vec{a} \cdot (W^l x_i^l \mid\mid w^l x_j^l)))}{\sum_{k \in N_i} exp(LeakyReLU(\vec{a} \cdot (W^l x_i^l \mid\mid W^l x_k^l)))} \tag{17}$$

Where $\vec{a}$ is the parameter vector of the forward layer, $||$ is the splicing operation, $W$ is a weight matrix, $N_i$ is the first-order neighborhood of node $i$, and $LeakyReLU$ is the activation function.

On the basis of calculating the normalized weight parameters of each node in each layer, the feature extraction process of each layer of the model can be expressed as:

$$H^l = \sigma(GAT(\alpha^{l-1} \cdot H^{l-1}, A)) \tag{18}$$

where $A$ is the adjacency matrix of nodes, and $H^{l-1}$ is the feature embedding vector matrix of each node generated in the previous layer. $\alpha^l$ is the normalized weight parameter generated by the previous layer.$H^k \in R^{n*d}$ is the node representation matrix of the $k$-th layer, $H^0 = X$. For the l-layer GAT, the activation function of the last layer is row-wise softmax prediction $Z = H^l$.

*(b)Observation model*

When using the Bayesian estimator to estimate the optimal graph, it is not enough to rely solely on the predicted embedding of the embedding module. In order to allow the structural model to observe more information to reduce estimation bias when estimating the graph structure, after $k$ iterations of aggregation, the embedded representation of the node captures the structural information within its k-order neighborhood, constructs observation maps from different angles based on the structural information of each order, and injects the observation model containing multi-order neighborhood similarity into the Bayesian inference model. , to provide node local to global information.

Specifically, construct the KNN graph $\{O_1, O_2 \ldots O_l\}$ as the observation graph model based on the feature matrix $H = \{H^1, H^2 \ldots H^l\}$ generated by each layer in the embedding model, where $O_i$ is The adjacency matrix of the kNN graph generated by $H^i$ represents the similarity of i-order neighborhoods. These generated KNN graphs reflect the best graph structures from different views and can be integrated to infer more reliable graph structures.

In particular, the expression generated by the observation map of each layer is:

$$u_{i,j} = \frac{x_i \cdot x_j}{|x_i| \, |x_j|} \tag{19}$$

Among them, $u$ is the similarity matrix of the node, which is obtained by finding the cosine similarity of the node vector. The final KNN proximity observation graph is formed by selecting the top $K$ nodes with the highest similarity for each node based on the similarity matrix.

*(c)Bayesian graphical structure estimator*

After generating the observation set $O$ and the predicted embedding $Z$, the next step is to derive a more realistic optimal graph S based on this information. So how can we generate the optimal estimation map? Although the observation sets describe the structure of the optimal graph from both local and global aspects, they are still insufficient and cannot be directly used as optimal estimation graphs. Therefore, in this work, the stochastic module (SBM) is first used to generate an optimal symmetric adjacency matrix with a community structure based on the prediction $Z$ and the label $Y$, and then the probability of mapping these observation sets O to this adjacency matrix is calculated, and finally Computational inversion is achieved by calculating the posterior distribution of the graph structure through Bayesian inference, thereby achieving the ultimate goal.

Specifically, a stochastic module (SBM) is first used to constrain the homogeneity of the generated graph structure by fitting the within-Community and between-community parameters in the block model to generate an estimated graph $Q$ with strong inter-community relationships. the estimated graph $Q$ is generated using the probability distribution $P(Q \mid \Omega, Z, Y)$, where $\Omega$ is a parameter of SBM, indicating the probability of edges linking within and between communities, for example, when belonging to groups $c_i$ and $c_j$ The probability of an edge between nodes i and j is $\Omega_{c_i c_j}$. The calculation to generate the estimated map $Q$ given the parameters $\Omega$, prediction $Z$ and label $Y$ can be expressed as:

$$P(Q \mid \Omega, Z, Y) = \prod_{i<j} \Omega_{c_i c_j}^{Q_{ij}} (1 - \Omega_{c_i c_j}^{1-Q_{ij}}) \tag{20}$$

$$c_i = \begin{cases} y_i & if v_i \in V_l \\ z_i & otherwise \end{cases} \tag{21}$$

where $y_i$ is the label of the node and $z_i$ is the predicted embedding. The probability $\Omega$ of the connecting edge between nodes is calculated by using the real label instead of the group category of the node in the training set.

Specifically, the probability $\Omega_{c_i c_j}$ of the existence of an edge in communities $c_i$ and $c_j$ is computed by averaging the probabilities of each edge between all nodes in these two communities, as follows:

Specifically, the probability $\Omega_{c_i c_j}$ of the existence of an edge in communities $c_i$ and $c_j$ is computed by averaging the probabilities of each edge between all nodes in these two communities, as follows:

$$\Omega_{c_i c_j} = \begin{cases} \frac{\varphi_{c_i c_j}}{\varphi_{c_i} \varphi_{c_j}} & if c_i \neq c_j \\ \frac{2\varphi_{c_i c_j}}{\varphi_{c_i}(\varphi_{c_i}-1)} & otherwise \end{cases} \tag{22}$$

Where $\varphi_{c_i}$ is the number of nodes in the community $c_i$, $\varphi_{c_i c_j}$ represents the sum of the probabilities of edges between nodes in communities $c_i$ and $c_j$.

Then, in order to improve the accuracy of the estimated graph, the structure of the graph must be inferred using as many external observation data as possible. Therefore, in this work, an observation model has also been introduced to describe how the estimated graph Q is mapped to the observation graph. It is assumed that the observed values of edges are independent and identically distributed Bernoulli random variables conditioned on whether the edge exists in the optimal graph. $P(O \mid Q, \delta, \omega)$ is the probability of observing the value $O$ given the estimated graph, parameter model $\delta$ and $\omega$, where $\delta$ represents the probability that an edge actually exists in the estimated graph, and $\omega$ represents the probability of an edge in the estimated graph $S$ The probability that no edge is observed in .

$$P(O \mid Q, \delta, \omega) = \prod_{i<j} \left[\delta^{E_{i,j}}(1-\delta)^{M-E_{i,j}}\right]^{Q_{i,j}} \times \left[\omega^{E_{i,j}}(1-\omega)^{M-E_{i,j}}\right]^{1-Q_{i,j}} \tag{23}$$

Where $M$ is the number of observations, $E_{i,j}$ is the number of edges observed in the observations, and $M - E_{i,j}$ is the number of edges not observed in the observations.

It is difficult to directly calculate the posterior probability distribution of the optimal graph, so the Bayesian inference method will be used to determine the posterior probability distribution of the estimated graph Q. The expression of Bayesian inference is as follows:

$$P(Q, \Omega, \delta, \omega \mid O, Z, Y) = \frac{P(O \mid Q, \delta, \omega)P(Q \mid \Omega, Z, Y)P(\omega)P(\delta)P(\Omega)}{P(O, Z, Y)} \tag{24}$$

It is assumed that the parameters are independent of each other. The posterior probability equations for the parameters $\Omega$, $\delta$, and $\omega$ can be obtained by summing over all possible values of the estimated graph $Q$

$$P(\Omega, \delta, \omega \mid O, Z, Y) = \sum_A P(Q, \Omega, \delta, \omega \mid O, Z, Y) \tag{25}$$

Maximizing the three parameters of the posterior probability $\Omega$, $\delta$, $\omega$ of Equation 17 will provide the maximum a posteriori estimate, based on the maximum a posteriori estimate, the adjacency matrix $S$ of the graph $Q$ can be estimated.

$$S_{i,j} = \sum_A q(Q)Q_{i,j} \tag{26}$$

The adjacency matrix $S$ indicates the possibility of an edge between a node and all its adjacent nodes, and $S_{i,j}$ indicates that the posterior probability value of the edge between node $i$ and node $j$ is between [0,1].

*(d) Iterative update* To update the optimally estimated symmetric adjacency matrix S, Equation 17 is maximized using the expectation-maximization (EM) algorithm [54]. Since Equation 17 is difficult to solve directly, it is solved in this work by applying Jensen's inequality to Equation 17:

$$logP(\Omega, \delta, \omega \mid O, Z, Y) \geq \sum_Q q(Q)log\frac{P(Q, \Omega, \delta, \omega \mid O, Z, Y)}{q(Q)} \tag{27}$$

where $q(Q)$ is the probability distribution on the estimated graph $Q$, and $\sum_Q q(Q) = 1$.

The maximum value is obtained when both sides of inequality 19 are equal, namely:

$$q(Q) = \frac{P(Q, \Omega, \delta, \omega \mid O, Z, Y)}{\sum_Q P(Q, \Omega, \delta, \omega \mid O, Z, Y)} \tag{28}$$

Finally, by applying Bayes' theorem and maximizing the posterior based on the EM algorithm, the expectation of the graph structure is finally obtained:

$$S_{i,j} = \frac{\Omega_{c_i,c_j}\delta^{E_{i,j}}(1-\delta)^{M-E_{i,j}}}{\Omega_{c_i,c_j}\delta^{E_{i,j}}(1-\delta)^{M-E_{i,j}} + (1+\Omega_{c_i,c_j})\omega^{E_{i,j}}(1-\omega)^{M-E_{i,j}}} \tag{29}$$

Calculation of the posterior probability distribution $q(Q)$ may be simplified by the value of the symmetric adjacency matrix $S_{i,j}$:

$$q(Q) = \prod_{i<j} S_{i,j}^{Q_{i,j}}(1-S_{i,j})1 - Q_{i,j} \tag{30}$$

It doesn't make much sense to save all the edges and the calculation is very heavy. Therefore, in this paper, a threshold $\varepsilon$ is set to screen out those edges smaller than $\varepsilon$, so as to obtain a symmetric critical matrix $S^Q$.

$$S_{i,j}^Q = \begin{cases} S_{i,j} & if Q_{i,j} > \varepsilon \\ 0 & otherwise \end{cases} \tag{31}$$

## A.3 BASELINES

By analyzing the comparison of baseline methods with similar work, the EGLNN method is compared to a number of other methods, including: (1) some traditional algorithms (e.g. DeepWalk, GCN) (2) some latest algorithms for graph structure learning (e.g. GEN) (3) some algorithms for anomalous node detection in IoT, blockchain (e.g. trans2vec AMBGAT)) (4) Some algorithms for knowledge distillation (e.g. DistillGCN, T2-GNN).

• DeepWalk [28] : DeepWalk is an algorithm for learning graph structure embeddings by sampling a sequence of nodes through a random walk and then mapping it into a low-dimensional vector space.

• GCN [18] : GCN is a deep learning model for graph data that learns node representations by aggregating node neighbor information.

• GEN [54] : GEN is a graph structure learning algorithm. By constructing and utilizing multiple types of information, a more realistic graph structure can be estimated.

• trans2vec [47] : trans2vec is a method for detecting blockchain phishing scams by mining transaction records in the Ethereum network to map the network structure into low-dimensional embedding vectors.

• AMBGAT[13]: AMBGAT is an algorithm to secure data in IoT by augmenting node features with attentional power and estimating graph topologies that conform to basic facts using graph structure learning.

• DistillGCN [38]: DistillGCN is the first method to extract knowledge from pre-trained GCN models, enabling knowledge transfer from teacher GCNs to students.

• T2-GNN [53]: T2-GNN is an approach to avoid interference between features and structures by designing feature-level and structure-level teacher models separately to provide targeted guidance to student models (base GNNs, e.g., GCNs) through distillation.

## A.4 IMPLEMENTATION DETAILS

The implementation of this model consists of a teacher model and a student model. Among them, in the teacher model, the GAT network is used, and each layer includes $K = 4$ attention head calculations. The model is trained using a 0.01 learning rate, a 5e-5 weight decay, a 50% dropout rate per layer, and an Adam optimizer.

Additionally, the dimensions for the Bayesian graph structure estimator and the hidden layer embeddings in the teacher model are selected from the set $\{64, 128, 256, 512, 768\}$. The embedding dimensions for the output layer of the teacher model are chosen from the set $\{2, 4, 8, 16, 32, 64\}$. Feature map $K \in \{2, \cdots, 10\}$, threshold $\varepsilon \in \{0.1 \cdots 0.9\}$. The number of iterations for teacher model optimization is set to 400. In the student model, the MLP model used is two-layer, the embedding dimension of the hidden layer is selected from $\{128, 512\}$, the distillation temperature parameter $T \in \{1, 2, \cdots, 5\}$, the balance parameter $\lambda \in \{0.1, 0.2, \cdots, 0.9\}$. During model training and testing, only the classification performance of labeled nodes was taken into account, and the parameters with the highest performance were preserved for testing. Five independent experiments utilizing distinct random seeds were conducted for each technique, and the average accuracy (Pre), F1 score, and recall were reported to evaluate the performance of the models.

