# OpenReview forum: "EGLNN:ENHANCED GRAPHLESS NEURAL NETWORK FOR IOT DATA STORAGE TRANSACTION SECURITY"
_ICLR.cc/2025/Conference — Submitted to ICLR 2025_

### Official Review · Reviewer_eqMh · 2024-10-31

**Soundness:** 1
**Presentation:** 2
**Contribution:** 2
**Rating:** 3
**Confidence:** 4

**Summary:**

EGLNN is a graphless neural network model designed for the Ethereum network, particularly for IoT data storage and transaction security. This model leverages knowledge distillation to create a smaller, more efficient student model that mimics a larger teacher GNN, which is challenging to deploy directly in IoT applications. Bayesian methods improve graph structure estimation, optimizing for real-world latency constraints in anomaly detection.

**Strengths:**

1、The paper is well-structured, comprehensive, and easy to read.
2、Knowledge distillation significantly reduces model size, making it suitable for deployment in memory-limited industrial environments.
3、The experimental results demonstrate the effectiveness of the proposed method.

**Weaknesses:**

1、The framework and experimental diagrams, along with the experimental tables, lack clarity and require optimization, as well as additional labeling for better understanding.
2、The paper lacks innovation, appearing as a fusion of multiple existing technologies rather than presenting original contributions.
3、Some of the citations in the article are problematic and require further verification and optimization.
4、The author cites a few existing Ethereum anomaly detection algorithms in the related work; however, the lack of comparative experiments with these algorithms diminishes the persuasiveness of the experimental results.
5、In related work (line 150), the athor mentions AMBGAT algorithm and cites it with paper [1]. However, upon reviewing the literature, I found that this algorithm actually appears in paper [2], which raises questions about the author's research rigour.
[1] EGNN: Graph structure learning based on evolutionary computation helps more in graph neural networks
[2] Adaptive multi-channel Bayesian graph attention network for IoT transaction security

**Questions:**

Please see the weaknesses section.

---

### Official Review · Reviewer_Eu5m · 2024-11-01

**Soundness:** 2
**Presentation:** 3
**Contribution:** 2
**Rating:** 5
**Confidence:** 4

**Summary:**

This paper proposes an Enhanced Graph-Less Neural Network (EGLNN) model to address the security issues in industrial IoT data storage and transaction scenarios. EGLNN is designed to overcome the limitations of traditional GNNs, which are slow and resource-intensive due to their reliance on message-passing principles. The model uses a knowledge distillation approach to transfer the knowledge learned by a large GNN model to a smaller MLP model, enabling faster inference and reduced resource requirements.
The proposed framework consists of two modules: a teacher GNN module and a student MLP module. The teacher GNN learns structural feature embeddings (SFE) of the optimal graph, which are then distilled to the student MLP. The student MLP generates embeddings similar to the SFE vector using only the node features of the original graph.
Experimental results show that EGLNN significantly improves performance, especially in data-scarce scenarios. It achieves higher accuracy than traditional knowledge distillation methods and is more suitable for industrial anomaly detection tasks due to its efficiency and scalability.

**Strengths:**

In the paper, authors suggest a two modules approach to address the anomaly detection tasks in a restrained IoT scenarios. The possible strengths of the so proposed method could be:
1. EGLNN's knowledge distillation approach significantly reduces model size and inference time, making it highly scalable for industrial applications.
2. he use of Bayesian reasoning to optimize the graph structure enhances the model's ability to learn and represent realistic topological information.
3. Experiments results suggest EGLNN outperforms state-of-the-art methods in anomaly detection tasks, especially in scenarios with limited training data.

**Weaknesses:**

1. The results fail to demonstrate the ressource part of EGLNN versus a GNN solution (i.e the performance of EGLNN-T is better than EGLNN). It supposes to be the main motivation to suggest a two modules approach.
2. The description of the dataset is not enough for reviewers to have a clear idea.
3. The current implementation of EGLNN is focused on static graphs and may not be well-suited for dynamic networks containing time-series information, which is common in industrial IoT applications.

**Questions:**

1.In section 4, the explanation of dataset ref to 3.2. It was not helpful to understand the dataset. Maybe an example should be added to clarify and help the reviewer to better evaluate the results.

2. In Fig 2., it is obvious that EGLNN-T achieved a net better performance than EGLNN (a typo here just below Fig2, should be Fig.2 instead of Fig.4). It is normal, but the explanation in (2) fail to defend the use of student model.

3. In fact which analysis in Section 4 stated clearly the advantage of using student model facing the resource constrains? I think just to say MLP is simple and efficient is not enough, the experimental result should prove it!

4. It seems that PE vector is very helpful for the improvement of the performance of the method. However, there is not enough explanation about why it could work that much, especially when we look at the ablation studies.

---

### Official Review · Reviewer_7Gy8 · 2024-11-02

**Soundness:** 2
**Presentation:** 1
**Contribution:** 2
**Rating:** 3
**Confidence:** 4

**Summary:**

This paper introduces the Enhanced Graphless Neural Network, a knowledge-distillation framework designed to detect anomalies on Ethereum platforms. The model leverages MLP to distill structural insights from a teacher GNN that has been trained on an enhanced graph. This MLP is combined with another MLP trained on node features. Together, these models provide improved performance over the teacher GNN while maintaining the faster inference necessary for industrial settings.

**Strengths:**

The proposed approach offers an interesting solution for anomaly detection by combining knowledge distillation and MLPs to optimize performance. The design of this framework can lead to faster inference, making it valuable in real-world industrial applications.

**Weaknesses:**

The paper leverages positional encodings, which are well-studied in GNNs, particularly in Graph Transformers. However, I am not fully convinced about the method chosen to calculate the PE vectors, as the paper uses one-hot encoding for these vectors. This method does not intuitively capture local, relative, or global positions within the graph. Further insights on effective PE in graphs are discussed in [1].

Rampášek et al., "Recipe for a General, Powerful, Scalable Graph Transformer," NeurIPS 2022.

Several symbols and variables in the equations are undefined, making it difficult to follow the methodology. For example, the symbol Q in Equation 2 is used without definition.

The term "distillation loss" in Equation 4 seems misleading, as it combines distillation with a form of local supervision loss.

MLP2 is trained solely on node attributes, contributing node-specific information without overfitting to graph structures. However, the paper does not clarify the advantage of this setup over a simpler pipeline with a single MLP distilling from a GNN trained on both positional encodings and node features.

The training process of MLP2, including the specific loss function used, is insufficiently explained. More details on the joint optimization of the two MLPs would improve clarity.

Some sections do not provide key information, making the methodology challenging to follow. For example, Section 4.1.1 mentions that datasets are "collected in subsection 3.2," but subsection 3.2 does not provide any details about dataset collection. Similarly, the semi-supervised learning approach, which supposedly uses both labeled and unlabeled data, is not supported by a clear methodological explanation in the main text.

The rationale behind the selection of only 60, 70, and 100 nodes for training and 1,000 nodes for testing is unclear. Additionally, Section 4.5 references Figure 7, which is absent, making it challenging to verify the results reported. The paper claims that the proposed approach is suitable for large-scale graph data, yet Section 4.1 states that the dataset includes only 60, 80, and 100 nodes for training, with 1,000 nodes for testing. This setup does not align with large-scale graph settings. A clearer explanation is needed to describe how the full graph of 376,759 nodes and 1,048,576 edges was utilized in the experiment.

**Questions:**

The paper leverages positional encodings, which are well-studied in GNNs, particularly in Graph Transformers. However, I am not fully convinced about the method chosen to calculate the PE vectors, as the paper uses one-hot encoding for these vectors. This method does not intuitively capture local, relative, or global positions within the graph. Further insights on effective PE in graphs are discussed in [1]. What is the rationale for choosing the PE method used?

Rampášek et al., "Recipe for a General, Powerful, Scalable Graph Transformer," NeurIPS 2022.

Can you please clarify how MLP1 and MLP2 are jointly optimized?

It is not clear how the datasets are being prepared for the settings in this paper.  60, 80, and 100 nodes for training is really a small subset of the 376,759  nodes available for training.

As explained in the weaknesses, can you improve the flow of the paper, defining all used symbols appropriately?

---

### Official Review · Reviewer_bXjL · 2024-11-04

**Soundness:** 2
**Presentation:** 2
**Contribution:** 1
**Rating:** 3
**Confidence:** 3

**Summary:**

This paper addresses the security challenges posed by cybercrimes in the Ethereum network, which is increasingly used for secure IoT data management. While Graph Neural Networks (GNNs) have shown promise for detecting abnormal nodes, the authors identify that conventional GNNs, relying on message-passing, are computationally intensive and impractical for industrial applications due to their slow training times and large model sizes.

To overcome these limitations, the study proposes an innovative solution, the Enhanced Graph-Less Neural Network (EGLNN). EGLNN improves the estimation of graph structures using a Bayesian graph structure estimator, optimizing the network structure and reducing the computational burden of GNNs. By employing a knowledge distillation approach (faculty-student distillation), EGLNN offers a lightweight model that maintains efficacy, making it more suitable for industrial environments with limited storage and high-speed requirements. This study contributes to the advancement of secure and scalable network anomaly detection for IoT-based Ethereum applications.

**Strengths:**

S1 : Considers at IoT network and propose a knowledge distillation network with claim to provide faster inference.

**Weaknesses:**

- W1: It is not clear the significance of Ethereum network platform that constraints the paper to just study that dataset. This also adds that experimentation is missing across other datasets. At least some base datasets should have been included. Again with IoT network, what is the importance or added constraints because of that. It is not clear.

- W2: Related works seems incomplete, the papers [1-3] could have cited  and compared against given the similar problem statement.

- W3: Source code is missing to verify the reproducibility of the work.


[1] Lu, Weigang, et al. "AdaGMLP: AdaBoosting GNN-to-MLP Knowledge Distillation." Proceedings of the 30th ACM SIGKDD Conference on Knowledge Discovery and Data Mining. 2024.

[2] Wu, Lirong, et al. "Teach Harder, Learn Poorer: Rethinking Hard Sample Distillation for GNN-to-MLP Knowledge Distillation." Proceedings of the 33rd ACM International Conference on Information and Knowledge Management. 2024.

[3] Wu, Lirong, et al. "Extracting low-/high-frequency knowledge from graph neural networks and injecting it into mlps: An effective gnn-to-mlp distillation framework." Proceedings of the AAAI Conference on Artificial Intelligence. Vol. 37. No. 9. 2023.

**Questions:**

- W4 / Q1: What was the consideration for D1, D2 and D3. We understand the sizes are different. But multiple variations of D1, D2 and D3 could have been tried to make claims stronger and average could have been taken.

- W5 / Q2 : Is the Ethereum network platform supposed to be dynamic? In case of that, were there any considerations made accordingly in methodology and experimentation?

- W6 / Q3: Use of Positional embedding, does it cause issue with the permutation invariance expected out of GNNs; I do not see any experiments targeting that.

---

### Meta-Review · Area_Chair_o854 · 2024-12-14

**Metareview:**

The paper proposes the Enhanced Graph-Less Neural Network (EGLNN) to significantly reduce model size and inference time, making it scalable for IOT devices with Ethereum. While all reviewers agree the proposed network is very efficient in the inference procedure, there are still some major concerns. First, the paper needs to add some discussions with some missing related work (listed in the reviews) and more details both in the methodology and the experiments sections. The details are important to correctly evaluate the proposed method's novelty and make the results reproducible. Second, the paper needs some careful proofreading and revision in the methodology and related work sections. Third, large dataset results are needed to verify the proposed method's setting in the IOT devices with Ethereum.

**Additional Comments On Reviewer Discussion:**

The author didn't provide a rebuttal so there is no author-review discussion.

---

### Decision · Program_Chairs · 2025-01-22

Reject